# Estimating graphical models for count data with applications to single-cell gene network

**Feiyi Xiao** *
School of Mathematical Sciences
Peking University
xiaofeiyi1217@pku.edu.cn

**Junjie Tang**\*
School of Mathematical Sciences
Peking University
junjie.tang@pku.edu.cn

**Huaying Fang**
Beijing Advanced Innovation Center for Imaging Theory and Technology
Capital Normal University
huayingfang@hotmail.com

**Ruibin Xi**
School of Mathematical Sciences
Center for Statistical Science
Peking University
ruibinxi@math.pku.edu.cn

## Abstract

Graphical models such as Gaussian graphical models have been widely applied for direct interaction inference in many different areas. In many modern applications, such as single-cell RNA sequencing (scRNA-seq) studies, the observed data are counts and often contain many small counts. Traditional graphical models for continuous data are inappropriate for network inference of count data. We consider the Poisson log-normal (PLN) graphical model for count data and the precision matrix of the latent normal distribution represents the network. We propose a two-step method PLNet to estimate the precision matrix. PLNet first estimates the latent covariance matrix using the maximum marginal likelihood estimator (MMLE) and then estimates the precision matrix by minimizing the lasso-penalized D-trace loss function. We establish the convergence rate of the MMLE of the covariance matrix and further establish the convergence rate and the sign consistency of the proposed PLNet estimator of the precision matrix in the high dimensional setting. Importantly, although the PLN model is not sub-Gaussian, we show that the PLNet estimator is consistent even if the model dimension goes to infinity exponentially as the sample size increases. The performance of PLNet is evaluated and compared with available methods using simulation and gene regulatory network analysis of real scRNA-seq data.

## 1 Introduction

Gaussian graphical model (GGM) [15, 28] is widely used for understanding the complex interactions of the observed variables in various fields [26, 14, 23]. GGM assumes that each sample is drawn from a multivariate Gaussian distribution, in which the precision matrix (i.e., the inverse of the covariance matrix) represents the network. In GGM, two nodes are connected if the corresponding

---

*Equal contribution. Corresponding author: Ruibin Xi (ruibinxi@math.pku.edu.cn)

36th Conference on Neural Information Processing Systems (NeurIPS 2022).

element in the precision matrix is non-zero. Based on the sparsity assumption of the precision matrix, researchers have proposed many methods for estimating the precision matrix of the GGM and established consistency theories, such as methods by maximizing the penalized log-likelihood [28, 7], by solving an equivalent regression problem with the lasso penalty [15, 18] or by minimizing a smooth convex loss function (called D-trace loss) with the lasso penalty [30]. Due to its high interpretability, well-established theoretical properties, and computational advantages, GGM has been adopted in many applications.

This paper is motivated by the gene regulatory network analysis based on the single-cell RNA sequencing (scRNA-seq) data. The recent breakthrough of scRNA-seq technologies has provided tremendous opportunities for understanding transcriptional states and activities and for uncovering gene regulatory networks at single cell level. However, gene expressions from scRNA-seq are count data and the Gaussian assumption is inappropriate. In particular, for the recent unique molecular identifier (UMI) based scRNA-seq data [6, 32], the expression counts contain many zeros. Transformations such as taking logarithms cannot make the Gaussian model a good approximation to the distribution of scRNA-seq expression data and may distort the correlation structure when there are many zeros in scRNA-seq data.

Poisson distribution is a natural choice for modeling count data. Researchers have generalized the univariate Poisson distribution to multivariate distributions for network analysis [9]. Some such models are the Poisson graphical model [2] and its variants generalized Poisson graphical models [25]. But these generalized Poisson graphical models can not handle the over-dispersion often observed in real data. Negative binomial distributions are often used to account for the over-dispersion [19], but it is more difficult to generalize negative binomial distributions to describe the network structure in the multivariate count data. Here, we use the multivariate Poisson log-normal (PLN) distribution for graphical modeling of count data. The PLN distribution is a hierarchical model of Poisson and multivariate log-normal distributions [1]. A random vector $\mathbf{Y} = (Y_1, \ldots, Y_p)^T$ is from a PLN distribution, if conditional on a latent vector $\mathbf{X} = (X_1, \ldots, X_p)^T$ with $(\log(X_1), \ldots, \log(X_p))^T \sim \mathrm{N}(\boldsymbol{\mu}, \Sigma)$, elements of Y are independent Poisson random variables with mean parameters $X_1, \ldots, X_p$.

The major advantage of the PLN model is that, similar to the GGM, the precision matrix $\Theta = \Sigma^{-1}$ can represent the network. Network recovery can be achieved by estimating $\Theta$. In addition, PLN distributions allow over-dispersion and hence are suitable for modeling count data with over-dispersion. In gene regulatory network analysis of scRNA-seq data, the PLN graphical model has a clear biological explanation. Previous experimental researches showed that gene expressions of single cells follow log-normal distributions [4]. The latent variables $X_1, \ldots, X_p$, representing the true expressions of $p$ genes in a single cell, can be reasonably assumed to be jointly log-normally distributed. The precision matrix of the log-normal distribution represents gene-gene interactions in single cells. The observed variables $\mathbf{Y}$ are measurements of the underlying true expressions $\mathbf{X}$. Largely speaking, the log-normal layer of the PLN model captures the biological fluctuation of gene expressions and the Poisson layer accounts for the technical and measurement noises. Only the biological fluctuation reflects gene-gene interactions and the regulatory network is the precision matrix of the latent log-normal model.

A few algorithms have been developed for estimating the precision matrix of the PLN model in high-dimensional settings. Compared with GGM, the likelihood of the PLN model is more complicated since it involves a high-dimensional integration and does not have a closed form. Maximizing the penalized log-likelihood of the PLN model is very difficult. Wu et al. [24] used Monte Carlo methods to approximate the log-likelihood and estimate the precision matrix by maximizing the penalized approximated log-likelihood of the PLN model. Chiquet et al. [5] developed a computationally more appealing method based on the variational approximation. However, these methods are all based on approximations of the log-likelihood and the accuracies of these approximations need to be further elaborated. More importantly, no convergence theory has been developed for these precision matrix estimators.

In this article, we propose a two-step approach named PLNet that first estimates the covariance matrix of the latent log-normal variables in the PLN model using the maximum marginal likelihood estimator (MMLE) and then estimates the sparse precision matrix by minimizing the lasso penalized D-trace loss [30]. One advantage of this two-step approach is that it avoids computing high-dimensional numerical integration that is involved in the log-likelihood of the multivariate PLN model. Minimizing the penalized D-trace loss is computationally cost-effective. Thus, this estimator

is generally computationally more efficient. Furthermore, we show that, under mild conditions, the estimator given by PLNet is a consistent estimator of the precision matrix in high dimensional settings. Comprehensive simulation studies show that PLNet provides more accurate estimates and is computationally more efficient than available methods. We also demonstrate the application of PLNet to scRNA-seq data.

## 2 Methods

### 2.1 Notation

For a column vector $\mathbf{a}$ with the $i$-th entry $a_i$, $\|\mathbf{a}\|_2 = (\mathbf{a}^T \mathbf{a})^{1/2}$ and $\|\mathbf{a}\|_1 = \sum_i |a_i|$. For a matrix $A$ with the $(i, j)$-th entry $A_{ij}$, $\|A\|_\infty = \max_{i,j} |A_{ij}|$ be the element-wise $l_\infty$-norm, $\|A\|_1 = \sum_{i,j} |A_{ij}|$ be the $l_1$-norm, $\|A\|_{1,\infty} = \max_i(\sum_j |A_{ij}|)$ be the $l_{1,\infty}$-norm, $\|A\|_F = \left(\sum_{i,j} |A_{ij}|^2\right)^{1/2}$ be the Frobenius norm, $\|A\|_2 = \max_{\|v\|_2=1} \|Av\|_2$ be the operator norm, $\|A\|_{1,\text{off}} = \sum_{i \neq j} |A_{ij}|$, $\|A\|_0 = \sum_{i,j} 1_{\{A_{ij} \neq 0\}}$ be the number of nonzero entries where $1_{\{\cdot\}}$ is the indicator function, $\text{tr}(A) = \sum_i A_{ii}$ be the trace, and $A \succeq 0$ means $A$ is positive semi-definite.

### 2.2 The PLN graphical model

Let $Y = [Y_{ij}]_{1 \leq i \leq n, 1 \leq j \leq p}$ be the observed count matrix, $\mathbf{Y}_i = (Y_{i1}, \ldots, Y_{ip})^T$ be the observed count data of the $i$th sample, and $\mathbf{X}_i = (X_{i1}, \ldots, X_{ip})^T$ be the latent random vector. In scRNA-seq data, $Y_{ij}$ and $X_{ij}$ are the observed expression and the underlying "true" expression of the $j$th gene in the $i$th cell, respectively. We assume that conditional on $\mathbf{X}_i$, elements of $\mathbf{Y}_i$ are independent Poisson random variables with mean $S_i X_{ij}$ ($1 \leq j \leq p$), where $S_i$ is a scaling factor sampled from one distribution $g(S)$ and we define $\mathbf{S} = (S_1, \ldots, S_n)^T$ as the scaling vector. In scRNA-seq data, $S_i$ corresponds to the library size of the $i$th cell. The library size is related to the total sequencing reads and can be estimated by the sum of counts within each cell or other methods [12, 13, 21]. The $\log(\mathbf{X}_i) = (\log(X_{i1}), \ldots, \log(X_{ip}))^T$ follows a multivariate normal random vector with mean $\boldsymbol{\mu}^*$ and covariance $\Sigma^*$. The precision matrix $\Theta^* = (\Sigma^*)^{-1}$ represents the network. In summary, we have the following graphical model for count data, for $1 \leq i \leq n$,

$$
\begin{aligned}
\mathbf{Y}_i | \mathbf{X}_i &\sim \prod_{j=1}^p \text{Poisson}(S_i X_{ij}), \\
\log(\mathbf{X}_i) &\sim \text{N}\left(\boldsymbol{\mu}^*, (\Theta^*)^{-1}\right).
\end{aligned}
\tag{1}
$$

We denote this PLN distribution by $\mathbf{Y}_i \sim \text{PLN}(S_i; \boldsymbol{\mu}^*, \Sigma^*)$. The above PLN model is a little different from the classical form [1], in which $S_i = 1$ for $1 \leq i \leq n$. We assume that the network $\Theta^*$ is sparse and therefore we could maximize the penalized log-likelihood to estimate $\Theta^*$. However, the likelihood function in the PLN model involves a $p$-dimensional integration and is difficult to be computed, especially when $p$ is large. Chiquet et al. developed a variational algorithm called VPLN to maximize the penalized log-likelihood [5]. Although the variational method is computationally more feasible than directly maximizing the penalized log-likelihood, the estimator's theoretical properties are difficult to be obtained.

We develop a two-step estimator called PLNet that is computationally efficient and has good theoretical properties. We first use the MMLE method to derive an estimator $\widehat{\Sigma}$ for $\Sigma^*$. Then, we apply the D-trace method [30] with the covariance estimator $\widehat{\Sigma}$ to estimate the sparse precision matrix $\Theta^*$. We show that the derived estimator $\widehat{\Theta}$ is a consistent estimator of $\Theta^*$ even when the model dimension goes to infinity exponentially as the sample size increases.

### 2.3 The maximum marginal likelihood estimator

We estimate $\boldsymbol{\mu}^* = [\mu_j^*]_{1 \leq j \leq p}$ and $\Sigma^* = [\sigma_{jk}^*]_{1 \leq j, k \leq p}$ by maximizing the marginal log-likelihood to avoid high dimensional integrations. Let $\mathbf{Y}_{\cdot j}$ be the $j$th column of $Y$. For $1 \leq j \leq p$, we estimate $\mu_j^*$ and $\sigma_{jj}^*$ by maximizing the marginal log-likelihood of $\mathbf{Y}_{\cdot j}$ and denote the estimator as $\tilde{\mu}_j$ and $\tilde{\sigma}_{jj}$, and we estimate $\sigma_{jk}^*$ by maximizing the marginal log-likelihood of $(\mathbf{Y}_{\cdot j}, \mathbf{Y}_{\cdot k})$ and

denote the estimator as $\tilde{\sigma}_{jk}$. The specific forms of these marginal log-likelihoods and optimization problems are shown in the Supplemental Material (S2.1). Further, we define the MMLE for $\boldsymbol{\mu}^*$, $\Sigma^*$ as $\widetilde{\boldsymbol{\mu}} = [\tilde{\mu}_j]_{1 \leq j \leq p}$, $\widetilde{\Sigma} = [\tilde{\sigma}_{jk}]_{1 \leq j,k \leq p}$. We apply the Newton-Raphson algorithm to maximize the marginal log-likelihood with the initial values taken as the moment estimator $\widetilde{\boldsymbol{\mu}}^m$ and $\widetilde{\Sigma}^m$ in the Supplemental Material (S1.1). The explicit forms of the first and second-order partial derivatives of the marginal log-likelihood function are shown in the Supplemental Material (S2.1). The integrations involved in optimizations can be well approximated by the method described in [1, 17].

The above MMLE $\widetilde{\Sigma}$ may be not positive semi-definite. However, the D-trace method requires the input covariance matrix estimator to be positive semi-definite to guarantee the convexity of the loss function. We project $\widetilde{\Sigma}$ to the space of positive semi-definite matrices and identify $\check{\Sigma}$ that is closest to $\widetilde{\Sigma}$ in the space, i.e.,

$$\check{\Sigma} = \arg\min_{A \succeq 0} \left\| A - \widetilde{\Sigma} \right\|_\infty. \tag{2}$$

Then we add a small positive definite matrix to $\check{\Sigma}$ and get a positive-definite estimator $\widehat{\Sigma}$ of $\Sigma^*$,

$$\widehat{\Sigma} = \check{\Sigma} + \left\| \check{\Sigma} - \widetilde{\Sigma} \right\|_\infty I, \tag{3}$$

where $I$ is the identity matrix. The optimization problem (2) for $\check{\Sigma}$ can be solved by a splitting conic solver [8].

With the covariance matrix estimator $\widehat{\Sigma}$, we apply the D-trace method to estimate the precision matrix,

$$\widehat{\Theta} = \arg\min_{\Theta \succeq 0} \frac{1}{2} \text{tr} \left( \widehat{\Sigma} \Theta^2 \right) - \text{tr}(\Theta) + \lambda_n \left\| \Theta \right\|_{1,\text{off}}. \tag{4}$$

The above optimization problem (4) can be solved by an alternating direction method of multipliers [30]. In this paper, we use a more efficient algorithm developed in [22] to calculate $\widehat{\Theta}$. We can also incorporate the prior information about the gene regulatory relationship to improve the performance of network inference. Let $E$ be the set of gene pairs that cannot have direct interactions from the prior information, we can incorporate the prior information $E$ by considering the following constrained optimization problem,

$$\widehat{\Theta} = \arg\min_{\Theta \succeq 0} \left\{ \frac{1}{2} \text{tr} \left( \widehat{\Sigma} \Theta^2 \right) - \text{tr}(\Theta) + \lambda_n \left\| \Theta \right\|_{1,\text{off}} : \Theta_{ij} = 0 \text{ for } (i,j) \in E \right\}.$$

The tuning parameter $\lambda_n$ is selected by minimizing the following approximate Bayesian information criterion (BIC) [31],

$$\left\| \frac{1}{2} (\widehat{\Theta}\widehat{\Sigma} + \widehat{\Sigma}\widehat{\Theta}) - I \right\|_F + \left( \|\widehat{\Theta}\|_0 \log n \right) / n. \tag{5}$$

## 3 Theoretical properties

In this section, we establish the theoretical properties in the high dimensional setting. We first prove that $\widehat{\Sigma}$ is a consistent estimator of $\Sigma^*$. The convergence rate for $\widehat{\Sigma}$ is similar to that of the sample covariance matrix for random variables with sub-Gaussian distribution. Then, under the same irrepresentability condition of the D-trace method in [30], we derive the edge recovery property and consistency for the PLNet estimator $\widehat{\Theta}$.

### 3.1 Notation

Let $G = \{(i,j) | \Theta_{ij}^* \neq 0\}$ be positions of non-zero elements in $\Theta^*$, $G^c$ be the complement set of $G$, $d$ be the maximum node degree in $\Theta^*$, $s = \|\Theta^*\|_0$, and $\theta_{\min} = \min_{(i,j) \in G} |\Theta_{ij}^*|$ be the minimal absolute value of nonzero elements of $\Theta^*$. Let $\lambda_{\max}(A)$ and $\lambda_{\min}(A)$ be the largest and smallest eigenvalues of a symmetric matrix $A$. We define $\Gamma^* = \Gamma(\Sigma^*) = (\Sigma^* \otimes I + I \otimes \Sigma^*)/2$, $\widehat{\Gamma} = \Gamma\left(\widehat{\Sigma}\right)$, where $A \otimes B$ is the Kronecker product. For a $p \times p$ matrix $A$, the row $(i-1)p + j$ and column $(k-1)p + l$ of $\Gamma(A)$ is $\Gamma(A)_{(i,j),(k,l)} = \left( A_{ik} 1_{\{j=l\}} + A_{jl} 1_{\{i=k\}} \right)/2$. For two subsets $T_1$ and $T_2$

of $\{(i,j)\,|1 \leq i, j \leq p\}$, we define $\Gamma(A)_{T_1,T_2}$ be the submatrix of $\Gamma(A)$ whose rows and columns indexed by $T_1$ and $T_2$, respectively. Other notations are as follows,

$$\gamma = 1 - \max_{(i,j) \in G^c} \left\| \Gamma^*_{(i,j),G} (\Gamma^*_{G,G})^{-1} \right\|_1,$$

$$k_\Gamma = \left\| (\Gamma^*_{G,G})^{-1} \right\|_{1,\infty}, k_\Sigma = \|\Sigma^*\|_{1,\infty}.$$

### 3.2 The irrepresentability condition and rate of convergence

We first present the necessary irrepresentability condition for establishing the rate of convergence for the estimator in PLNet. This irrepresentability condition is from the D-trace method in [30].

**Condition 1 (Irrepresentability condition)** *There exists $0 < \alpha < 1$, such that*

$$\max_{(i,j) \in G^c} \left\| \Gamma^*_{(i,j),G} (\Gamma^*_{G,G})^{-1} \right\|_1 \leq 1 - \alpha.$$

The irrepresentability condition 1 is equivalent to $\gamma \geq \alpha > 0$. We also need a boundedness condition in the PLN model (1).

**Condition 2 (Boundedness condition)** *There exist positive constants $M_1, M_2, M_3, M_4$, and $M_5$, such that the distribution of $S_i$ has bounded support $[M_1, M_2]$, $\max_{1 \leq j,k \leq p} \left\{ \left| \mu_j^* \right|, \left| \sigma_{jk}^* \right| \right\} \leq M_3$ and $M_4 \leq \lambda_{min}(\Sigma^*) \leq \lambda_{max}(\Sigma^*) \leq M_5$.*

Based on the boundedness condition 2, we can establish the convergence rate for the covariance matrix estimator $\widehat{\Sigma}$.

**Theorem 1 (Rate of convergence for $\widehat{\Sigma}$)** *Under Condition 2, there exist positive constants $A, B$, such that for any $\epsilon > 0$,*

$$pr\left( \left\| \widehat{\Sigma} - \Sigma^* \right\|_\infty \geq \epsilon \right) \leq p^2 A \exp(-Bn\epsilon^2).$$

After plugging $\widehat{\Sigma}$ into the lasso penalized D-trace loss, we get a consistent estimator $\widehat{\Theta}$ that converges to $\Theta^*$ in several matrix norms.

**Theorem 2 (Rate of convergence for $\widehat{\Theta}$)** *Under the irrepresentability condition 1 and the boundedness condition 2, there exist constants $A, B$, such that for some $\eta > 2$, if*

$$n > \quad B^{-1}(\eta \log p + \log A) \max\left[ 12dk_\Gamma, \ 12\gamma^{-1}(k_\Sigma k_\Gamma^2 + k_\Gamma), \ \left\{ 12\gamma^{-1}\left( k_\Sigma k_\Gamma^3 + k_\Gamma^2 \right) + 5dk_\Gamma^2 \right\} \theta_{\min}^{-1}, \right.$$

$$\left. \min\left\{ s^{1/2}, d+1 \right\} \left\{ 12\gamma^{-1}\left( k_\Sigma k_\Gamma^3 + k_\Gamma^2 \right) + 5dk_\Gamma^2 \right\} \lambda_{\min}^{-1}(\Theta^*) \right]^2,$$

*and*

$$\lambda_n = 12\gamma^{-1}\left( k_\Sigma k_\Gamma^2 + k_\Gamma \right) B^{-1/2}(\eta \log p + \log A)^{1/2} n^{-1/2},$$

*then with probability $1 - p^{2-\eta}$,*

$$\left\| \widehat{\Theta} - \Theta^* \right\|_\infty \leq \left( 12\gamma^{-1}\left( k_\Sigma k_\Gamma^3 + k_\Gamma^2 \right) + 5dk_\Gamma^2 \right) B^{-1/2}(\eta \log p + \log A)^{1/2} n^{-1/2},$$

$$\left\| \widehat{\Theta} - \Theta^* \right\|_F \leq s^{1/2} \left( 12\gamma^{-1}\left( k_\Sigma k_\Gamma^3 + k_\Gamma^2 \right) + 5dk_\Gamma^2 \right) B^{-1/2}(\eta \log p + \log A)^{1/2} n^{-1/2},$$

$$\left\| \widehat{\Theta} - \Theta^* \right\|_2 \leq \min\left\{ s^{1/2}, d+1 \right\} \left( 12\gamma^{-1}\left( k_\Sigma k_\Gamma^3 + k_\Gamma^2 \right) + 5dk_\Gamma^2 \right) B^{-1/2}(\eta \log p + \log A)^{1/2} n^{-1/2}.$$

Meanwhile, with a high probability, the sign of the sparse precision matrix $\Theta^*$ can be recovered by $\widehat{\Theta}$. We have the following theorem about the sign consistency of $\widehat{\Theta}$.

**Theorem 3 (Sign consistency for $\widehat{\Theta}$)** *Under the conditions in Theorem 2, for some $\eta > 2$, choosing the same $n$ and $\lambda_n$ in Theorem 2, then with probability $1 - p^{2-\eta}$, $\widehat{\Theta}$ recovers all zeros and nonzeros in $\Theta^*$.*

The rate-of-convergence and sign consistency results in Theorem 2 and 3 are closely related to the tail probability of the MMLE estimator $\widehat{\Sigma}$ in Theorem 1. The boundedness condition 2 is assumed to guarantee the convergence of the MMLE. Ignoring the complicated constants in theorems, for any $\eta > 2$, if we have $n > C\eta \log p$, or in other words, if $p$ tends to infinity not faster than exponential of the sample size $n$, $\widehat{\Theta}$ is a consistent estimator of $\Theta^*$. Especially, the rate of convergence for $\widehat{\Theta}$ is $O\left([\eta(\log p)/n]^{1/2}\right)$ under $l_\infty$-norm.

## 4 Simulation studies

### 4.1 Simulation settings

We conduct simulations to evaluate the performance of PLNet and compare it with the available network inference methods including VPLN [5], glasso [7], and an estimator called PLNet-MOM by plugging the moment estimator $\widehat{\Sigma}^m$ into D-trace loss (See Supplemental Material (S1.1)). Both PLNet and VPLN are designed to estimate the precision matrix for count data in the PLN model. The glasso algorithm is a classical approach for continuous data in GGM and we apply glasso to the logarithmic transformation of the normalized data, which is defined as $\tilde{Y}_{ij} = (Y_{ij} + 1)/\hat{S}_i$, $1 \leq i \leq n$, where $\hat{S}_i$ is the estimated library size of $i$th cell. In all simulations, we estimate the library size by the total sum scaling (i.e. $\hat{S}_i = \sum_{j=1}^{p} Y_{ij}$), which is a classical normalization method for scRNA-seq and is defined as the sum of counts within each cell.

We consider 48 different simulation scenarios, which are 2 sample size setups ($n = 500, 2000$) $\times$ 3 dimension setups ($p = 100, 300, 500$) $\times$ 2 dropout levels (low and high) $\times$ 4 graph structures and simulate count data from the PLN model, where the dropout level represents the proportion of zeros in the count matrix. Details of the simulation settings are shown in the Supplementary Material (S3.1). In each scenario, we independently repeat simulations 100 times. The four graph structures are as follows.

1. Banded Graph: Pairs $(i, j)$ of nodes are connected if $|i - j| \leq 2$, $i \neq j$. All nonzero edges are set as $0.3$.

2. Random Graph: Pairs of nodes are connected with probability $0.1$. The nonzero edges are set as $0.3$ with probability $0.8$ and as $-0.3$ with probability $0.2$.

3. Scale-free Graph: The Barabasi-Albert model [3] is used to generate a scale-free graph with power 1. The nonzero edges are set as $0.3$.

4. Blocked Graph: The nodes are divided into 5 blocks of equal sizes. Pairs of nodes in the same block are connected with probability $0.1$ and the nonzero edges are set as $0.3$. Different blocks have no edge connection.

The diagonal elements of precision matrices are set as 1 plus a positive number to guarantee positive definiteness.

### 4.2 Performance comparison

Table 1 and Table 2 show the area under the precision and recall curve (AUPR) of each estimator. AUPRs are calculated by varying the tuning parameters (i.e. the penalty parameters of the four estimators). As expected, the AUPR decreases as the number of genes increases. AUPRs in the high-dropout cases are generally smaller than that in the low-dropout cases. PLNet is the most robust estimation among these four estimators and outperforms PLNet-MOM, VPLN, and glasso in most simulation settings in AUPR, especially for the settings with high dropouts. For example, for $n = 2000, p = 100$, PLNet achieves an AUPR of $0.88$ for the random graph under the scenario of the high dropout rate, while PLNet-MOM, VPLN, and glasso only have AUPRs of $0.82$, $0.69$, and $0.18$, respectively. Furthermore, PLNet-MOM outperforms glasso in most settings and outperforms

Table 1: Comparisons of PLNet, VPLN, glasso, and PLNet-MOM in terms of the area under precision and recall curve (AUPR) on simulation results for $n = 500$. The results are averages over 100 replicates with standard deviations in brackets.

| Sample size | $n = 500$ | | $n = 500$ | | $n = 500$ | |
|---|---|---|---|---|---|---|
| Dimension | $p = 100$ | | $p = 300$ | | $p = 500$ | |
| Dropout | Low | High | Low | High | Low | High |
| Banded graph | | | | | | |
| PLNet | **0.79 (0.03)** | **0.51 (0.04)** | **0.6 (0.06)** | **0.31 (0.04)** | 0.44 (0.08) | **0.22 (0.03)** |
| PLNet-MOM | 0.62 (0.03) | 0.43 (0.03) | 0.33 (0.02) | 0.21 (0.01) | 0.21 (0.02) | 0.13 (0.01) |
| VPLN | 0.68 (0.06) | 0.44 (0.04) | 0.6 (0.02) | 0.27 (0.02) | **0.54 (0.01)** | 0.21 (0.02) |
| glasso | 0.34 (0.04) | 0.05 (0.01) | 0.44 (0.03) | 0.04 (0.01) | 0.44 (0.02) | 0.04 (0.01) |
| Random graph | | | | | | |
| PLNet | **0.73 (0.07)** | **0.49 (0.07)** | **0.6 (0.07)** | 0.25 (0.05) | **0.53 (0.09)** | 0.16 (0.04) |
| PLNet-MOM | 0.6 (0.07) | 0.44 (0.07) | 0.44 (0.04) | 0.22 (0.04) | 0.36 (0.04) | 0.14 (0.03) |
| VPLN | 0.59 (0.07) | 0.44 (0.08) | 0.53 (0.09) | **0.26 (0.05)** | 0.51 (0.06) | **0.18 (0.05)** |
| glasso | 0.41 (0.05) | 0.14 (0.02) | 0.48 (0.05) | 0.11 (0.02) | 0.48 (0.05) | 0.09 (0.02) |
| Scale-free Graph | | | | | | |
| PLNet | **0.73 (0.12)** | **0.55 (0.08)** | **0.67 (0.04)** | **0.44 (0.05)** | **0.59 (0.04)** | 0.35 (0.06) |
| PLNet-MOM | 0.63 (0.08) | 0.51 (0.06) | 0.48 (0.03) | 0.34 (0.03) | 0.39 (0.01) | 0.27 (0.03) |
| VPLN | 0.67 (0.05) | 0.53 (0.09) | 0.61 (0.03) | 0.43 (0.04) | 0.57 (0.07) | **0.38 (0.05)** |
| glasso | 0.56 (0.06) | 0.34 (0.05) | 0.59 (0.02) | 0.32 (0.03) | 0.58 (0.02) | 0.3 (0.04) |
| Blocked graph | | | | | | |
| PLNet | **0.68 (0.05)** | **0.47 (0.08)** | **0.59 (0.08)** | **0.24 (0.07)** | **0.45 (0.1)** | **0.17 (0.04)** |
| PLNet-MOM | 0.58 (0.04) | 0.43 (0.08) | 0.43 (0.06) | 0.2 (0.05) | 0.32 (0.05) | 0.15 (0.03) |
| VPLN | 0.59 (0.04) | 0.45 (0.07) | 0.52 (0.07) | 0.24 (0.06) | 0.44 (0.08) | 0.17 (0.04) |
| glasso | 0.37 (0.03) | 0.16 (0.03) | 0.44 (0.06) | 0.11 (0.03) | 0.4 (0.06) | 0.09 (0.02) |

Table 2: Comparisons of PLNet, VPLN, glasso, and PLNet-MOM in terms of the area under precision and recall curve (AUPR) on simulation results for $n = 2000$. The results are averages over 100 replicates with standard deviations in brackets.

| Sample size | $n = 2000$ | | $n = 2000$ | | $n = 2000$ | |
|---|---|---|---|---|---|---|
| Dimension | $p = 100$ | | $p = 300$ | | $p = 500$ | |
| Dropout | Low | High | Low | High | Low | High |
| Banded graph | | | | | | |
| PLNet | **0.99 (0.01)** | **0.96 (0.01)** | **0.99 (0.01)** | **0.94 (0.02)** | **0.98 (0.01)** | **0.89 (0.08)** |
| PLNet-MOM | 0.97 (0.01) | 0.92 (0.01) | 0.91 (0.01) | 0.83 (0.02) | 0.83 (0.02) | 0.75 (0.01) |
| VPLN | 0.95 (0.01) | 0.89 (0.03) | 0.94 (0.01) | 0.79 (0.15) | 0.94 (0.01) | 0.81 (0.01) |
| glasso | 0.62 (0.03) | 0.04 (0.01) | 0.82 (0.01) | 0.07 (0.01) | 0.85 (0.01) | 0.15 (0.02) |
| Random graph | | | | | | |
| PLNet | **0.98 (0.01)** | **0.88 (0.04)** | **0.98 (0.03)** | **0.85 (0.05)** | **0.99 (0.01)** | **0.83 (0.04)** |
| PLNet-MOM | 0.94 (0.02) | 0.82 (0.06) | 0.94 (0.01) | 0.77 (0.05) | 0.93 (0.01) | 0.74 (0.05) |
| VPLN | 0.78 (0.08) | 0.69 (0.07) | 0.88 (0.03) | 0.67 (0.1) | 0.86 (0.11) | 0.67 (0.11) |
| glasso | 0.55 (0.06) | 0.18 (0.03) | 0.8 (0.03) | 0.24 (0.04) | 0.84 (0.02) | 0.26 (0.04) |
| Scale-free Graph | | | | | | |
| PLNet | **0.89 (0.17)** | **0.85 (0.11)** | **0.97 (0.02)** | **0.85 (0.03)** | **0.96 (0.03)** | **0.83 (0.02)** |
| PLNet-MOM | 0.85 (0.11) | 0.81 (0.08) | 0.86 (0.01) | 0.75 (0.02) | 0.83 (0.01) | 0.71 (0.01) |
| VPLN | 0.74 (0.16) | 0.67 (0.15) | 0.79 (0.04) | 0.68 (0.11) | 0.8 (0.05) | 0.66 (0.13) |
| glasso | 0.59 (0.14) | 0.45 (0.06) | 0.78 (0.02) | 0.5 (0.03) | 0.81 (0.02) | 0.53 (0.02) |
| Blocked graph | | | | | | |
| PLNet | **0.94 (0.02)** | **0.83 (0.07)** | **0.97 (0.01)** | **0.81 (0.08)** | **0.97 (0.01)** | **0.77 (0.05)** |
| PLNet-MOM | 0.88 (0.04) | 0.75 (0.08) | 0.91 (0.02) | 0.72 (0.08) | 0.89 (0.02) | 0.68 (0.05) |
| VPLN | 0.73 (0.03) | 0.66 (0.07) | 0.78 (0.04) | 0.62 (0.07) | 0.8 (0.06) | 0.59 (0.11) |
| glasso | 0.47 (0.05) | 0.2 (0.03) | 0.7 (0.04) | 0.21 (0.04) | 0.75 (0.03) | 0.21 (0.04) |

VPLN under the scenario of large sample sizes. The results of the area under the receiver operating characteristic curve (AUC) are similar and shown in Supplementary Table S1-S2.

To further demonstrate the performance of PLNet, we visualize the mean networks predicted by the four methods tuned by BIC criterion for the banded graph with $n = 2000, p = 100$ over the 100 simulations (Figure 1). More specifically, we calculate the relative frequency $F_{ij}$ that an algorithm reports edges over the 100 simulations. For positions $(i, j)$ with $\Theta_{ij} \neq 0$, $F_{ij}$ is the proportion that an algorithm correctly recovers the edge in the 100 simulations, while for positions $(i, j)$ with $\Theta_{ij} = 0$, $F_{ij}$ is the proportion that an algorithm falsely predicts edges between nodes $i$ and $j$ in the 100 simulations. We plot the relative frequency matrices of the four methods in Figure 1. We clearly see that PLNet is able to detect more true positives while having fewer false positives than other methods, especially for the high dropout case. The results of mean predicted networks for the other three graphs with $n = 2000, p = 100$ are similar and shown in Supplementary Figure S1-S3.

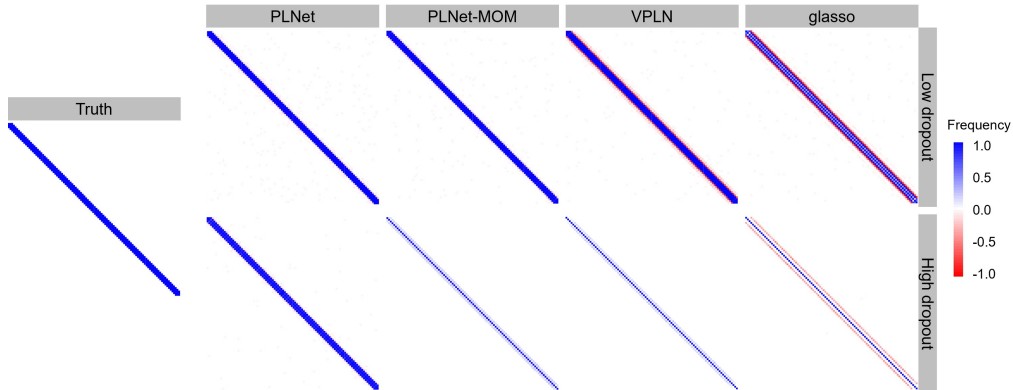

Figure 1: The mean networks predicted by PLNet, VPLN, glasso, and PLNet-MOM for the banded graph with 100 nodes and n = 2000. False edges are colored in red and true edges are in blue. The left panel is the true network matrix for reference.

Supplementary Table S3-S4 show the computational time of the four methods. PLNet is computationally more efficient than VPLN in most simulation settings. Also, the computational efficiency of PLNet is roughly the same for different sample sizes, while the computational complexity of VPLN maintains a linear relationship with sample size. The detailed comparisons between the computational efficiency of the four methods are shown in the Supplementary Material (S3.2.2).

## 5    Application to a scRNA-seq dataset

We apply PLNet and VPLN to infer the gene regulatory network of CD14+ Monocytes profiled in Kang et al. [10]. The single cells are profiled in two different conditions, IFN-$\beta$-treated and controlled. IFN-$\beta$ is a cytokine in the interferon family that influences the transcriptional profiles for many genes, especially those in the JAK/STAT pathway [16]. We focus on the IFN-$\beta$-treated cells (2147 cells) and use the top 200 highly variable genes that are used in [20] for network analysis.

We compare networks of PLNet and VPLN with parameters tuned such that the network densities are around 5%. Gene Ontology (GO) analysis [11] shows that the 200 genes mainly involve in 4 biological processes, including "Cytokine-mediated signaling pathway" (Module $M_1$), "Neutrophil-mediated immunity" (Module $M_2$), "Cellular protein metabolic process" (Module $M_3$), and "Proteolysis" (Module $M_4$). Figure 2 shows the predicted networks of the genes in the 4 modules by PLNet and VPLN, where the colors represent the partial correlations between genes. The partial correlation given by PLNet between genes $i$ and $j$ is defined as $-\widehat{\Theta}_{ij}/(\widehat{\Theta}_{ii}\widehat{\Theta}_{ij})^{1/2}$. The partial correlation given by VPLN is defined similarly. We clearly see that the network given by PLNet tends to have more connections within the modules than VPLN. To see this more clearly, for each module $M_k$, we calculate the ratio between within-module and between-module connections $R(M_k) = \Sigma_{i,j \in M_k} W_{ij}/\Sigma_{i \in M_k, j \notin M_k} W_{ij}$, where the weights $W_{ij}$ are set as the absolute partial correlation between genes $i$ and $j$ (weighted within-between connection ratio) or are set as 1 and 0 depending on whether genes $i$ and $j$ are connected (unweighted within-between connection ratio). The within-

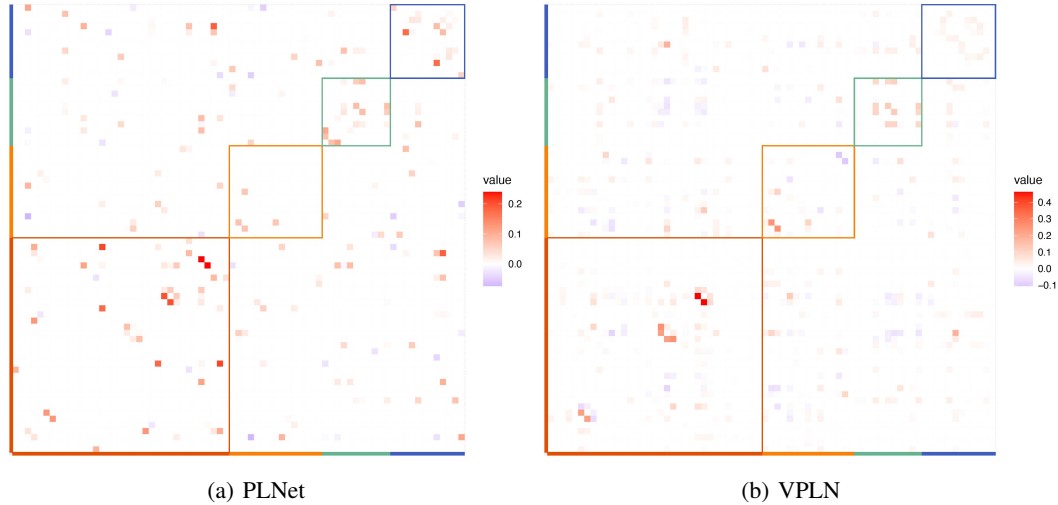

|  | (a) PLNet |  | (b) VPLN |
|--|-----------|--|----------|

Figure 2: Heat maps of partial correlations between genes in the 4 GO modules given by PLNet (a) and VPLN (b). Red: Cytokine-mediated signaling pathway (Module $M_1$); Orange: Neutrophil-mediated immunity (Module $M_2$); Green: Cellular protein metabolic process (Module $M_3$); Blue: Proteolysis (Module $M_4$).

between connection ratios of PLNet are much larger than that of VPLN in most cases (Table 3). Similar results also hold for networks of other different densities that are shown in the Supplementary Material (S4.1).

Table 3: The within-between connection ratios of the 4 modules in the networks estimated by PLNet and VPLN tuned such that the network densities are around 5%.

| Type | Method | Module M1 | Module M2 | Module M3 | Module M4 |
|------|--------|-----------|-----------|-----------|-----------|
| Weighted | PLNet | **0.681** | 0.231 | **0.582** | **0.387** |
|  | VPLN | 0.621 | **0.381** | 0.459 | 0.259 |
| Unweighted | PLNet | **0.533** | 0.129 | **0.419** | **0.383** |
|  | VPLN | 0.515 | **0.200** | 0.170 | 0.213 |

To further compare the performance of PLNet and VPLN on the real scRNA-seq data analysis, we construct a silver standard based on an available regulatory network database obtained from ChIP-seq experiments (the hTFtarget database [29]). The silver standard consists of edges between the 200 highly variable genes and 36 transcription factors (TF) from the top 500 highly variable genes. We consider 26 additional TFs from the top 500 highly variable genes because the top 200 highly variable genes only contain 10 TFs and the silver standard constructed only using these genes is too sparse to be a reliable silver standard. We compare networks estimated by PLNet and VPLN with their parameters tuned such that the network densities are at the same levels (1% to 10%). Table 4 shows the number of true edges in the silver standard detected by PLNet and VPLN at different network densities. Clearly, we see that PLNet detects more true edges at the same density levels than VPLN, suggesting that PLNet has a higher true discovery rate than VPLN.

Table 4: The number of true edges estimated by two methods with different density levels.

| Density | 0.01 | 0.02 | 0.03 | 0.04 | 0.05 | 0.06 | 0.07 | 0.08 | 0.09 | 0.10 |
|---------|------|------|------|------|------|------|------|------|------|------|
| PLNet | **8** | **16** | **23** | **35** | **41** | **44** | **62** | **73** | **81** | **92** |
| VPLN | 2 | 5 | 7 | 12 | 20 | 27 | 36 | 48 | 62 | 62 |

## 6 Discussion

In this paper, we consider the PLN graphical model for count data. This model has an intuitive explanation for single-cell gene regulatory network analysis. The Poisson layer is to capture the

technical noises and the log-normal layer is to model the biological fluctuations in single cells. Gene regulatory network is represented by the precision matrix of the latent log-normal model. To estimate the precision matrix, we propose a two-step estimator PLNet, using the MMLE to estimate the covariance matrix and then minimizing the penalized D-trace loss to estimate the precision matrix. The simplicity of this estimation procedure allows us to establish the consistency theory for the proposed PLNet estimator for the high dimensional setting. The numerical analysis also shows that the PLNet method outperforms available methods.

The proposed method can be generalized in several ways. First, we may consider the zero-inflated PLN (ZIPLN) model to account for the excessed number of zeros that cannot be modeled by PLN models. We could also use the MMLE method to estimate the covariance matrix of the ZIPLN model, but the convergence theory is more difficult to develop. Secondly, a straightforward generalization is the differential network analysis. We could use the same method to estimate covariance matrices in two different conditions and use the D-trace loss developed in earlier work [27] for differential network analysis in single cells. Thirdly, the mean parameters of the latent log-normal random variable can depend on covariates. In such model, we could first use regression to estimate the mean parameters and derive the corresponding moment estimator of the covariance matrix. Another generalization is gene regulatory network analysis of mixtures of cell populations. Different cell populations may have different gene regulatory networks and we could jointly model the mixture and infer the gene regulatory networks for all cell populations. Finally, the current method is based on parametric inference. If the model is mis-specified, network inference could be misleading. To mitigate this problem, we can consider a semi-parametric model by replacing the latent log-normal model with a Gaussian copula graphical model, but how to do network inference with this semi-parametric model is still a challenge that needs to be overcome in the future.

## Acknowledgments and Disclosure of Funding

We thank Changhu Wang and Siyuan Huang for helpful discussions. This work was supported by the National Natural Science Foundation of China [No. 11971039 to R.X.], the National Key Basic Research Program of China [2020YFE0204000 to R.X.], and Sino-Russian Mathematics Center.

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
