# OpenReview forum: "Estimating graphical models for count data with applications to single-cell gene network"
_NeurIPS.cc/2022/Conference — NeurIPS 2022 Accept_

### Official Review · Reviewer_gBop · 2022-07-09

**Rating:** 7
**Confidence:** 3
**Soundness:** 4 excellent
**Presentation:** 4 excellent
**Contribution:** 3 good

**Summary:**

The authors propose a new algorithm (PLNet) for estimating the precision matrix of a Poisson log-normal (PLN) graphical model for count data. The authors evaluate their method on both simulated and real-world data and find improvements over baseline methods.

**Questions:**

1. Before reviewing this manuscript, I was not familiar with the VPLN model. As such, it wasn't clear to me why PLNet is computationally more efficient than VPLN. It would be great if the authors could provide more details on this.

**Limitations:**

The authors addressed the limitations of their work in a satisfactory manner. They did not discuss potential negative societal impact, which is fine for this work (I do not see any potential negative impacts here).

**Strengths And Weaknesses:**

Overall I was quite happy with the paper and recommend acceptance.

Strengths:

1. The authors tackle a high-impact problem; rich scRNA-seq datasets are becoming more and more common, and methods for analyzing them are in high demand. Moreover, many works have found that analyzing scRNA-seq datasets after first normalizing and log-transforming the data leads to significant distortions. As such, methods that are able to operate on raw counts are crucial.
2. To my knowledge, the authors' proposed algorithm (as well as the theoretical analysis of it) is novel.
3. The authors demonstrate that their algorithm outperforms naive GGM approaches as well as a PLN approach that optimizes a lower bound to the log-likelihood (as opposed to directly optimizing the likelihood).
4. The writing in the manuscript was clear and I generally found the paper to be well-structured.

Weaknesses:

1. Ideally I would've liked to see another application on real-world data, though given space constraints I understand that may not have been possible to fit in.

---

> ### Author Response · Authors · 2022-08-02
> **Response to Reviewer gBop**
>
> Thank you for your positive feedback and helpful comments.
>
> For weaknesses:
> >"1. Ideally I would've liked to see another application on real-world data, though given space constraints I understand that may not have been possible to fit in."
>
> We totally agree with the reviewer that another application would be better to demonstrate that the method can be applied to infer networks of count observations from different areas. However, due to space constraints as well as rebuttal time constraints, we cannot have another application on real-world data in this paper.
>
> For questions:
> >"1. Before reviewing this manuscript, I was not familiar with the VPLN model. As such, it wasn't clear to me why PLNet is computationally more efficient than VPLN. It would be great if the authors could provide more details on this."
>
> The computational efficiency of PLNet over VPLN can be explained  in the following two aspects. First, since VPLN adopts the variational approach to estimate network, in addition to the $p(p+3)/2$ unknown parameters in the PLN model, VPLN has to estimate additional $2np$ variational parameters. When the sample size $n$ is large, the computational burden for these additional variational parameters is very high for VPLN. In comparison, PLNet only need to estimate the $p(p+3)/2$ parameters in the PLN model. Second, at each iteration of the VPLN updates, VPLN has to solve a Gaussian graphical lasso problem, and PLNet only needs to solve one penalized D-trace problem that is similar to the Gaussian graphical lasso problem in terms of the computational burden.

---

### Official Review · Reviewer_CQBu · 2022-07-11

**Rating:** 7
**Confidence:** 3
**Soundness:** 4 excellent
**Presentation:** 4 excellent
**Contribution:** 3 good

**Summary:**

This paper introduces a consistent estimator for the sparse precision matrix of the Poisson log-normal (PLN) graphical model. The paper provides theoretical guarantees of the estimator's consistency and convergence rate. Using simulated data, the paper demonstrates that the proposed estimator is more accurate than competitive baselines across a variety of conditions. The paper also compares the proposed estimator to a recent baseline on inferring a gene regulatory network from RNA sequence data.

**Questions:**

* In the experiments section there are 3 instances where PLNet and VPLN seem to achieve the same AUPR, but only PLNet is bolded. Is this just due to rounding?

* What is the relationship of the proposed method to the method given in section 2.4.2 of Fan, Liao, and Liu (2015) [1]?

[1] An Overview on the Estimation of Large Covariance and Precision Matrices. Jianqing Fan, Yuan Liao and Han Liu. 2015.
https://arxiv.org/pdf/1504.02995.pdf

**Limitations:**

Yes.

**Strengths And Weaknesses:**

## Clarity
This paper is very well-written and clear. It provides excellent coverage of prior work and situates the proposed approach well within the relevant literature. The technical exposition is cleanly structured and easy to follow. The experiments are reported with sufficient detail. I also appreciate that the proposed method is grounded in an applied problem, which provides strong motivation for the paper, and helps in understanding the method.

## Quality
I found no flaws in the paper's logic or in the proofs for theorems 1-3 (which I only skimmed). The proposed method is simple and practical, while the theory guarantees root-n convergence of the estimator under mild conditions.

The experiments are thorough. The paper does a good job of simulating many synthetic data sets under varying conditions and performance to reasonable baselines. One drawback is that only the simple case of a banded covariance matrix is given as an example to compare visually the results of the proposed method to the baselines. Perhaps the authors can add more visual inspections to the appendix.

Minor comments/edits:
- "semi-positive definite" -> "positive semi-definite"
- line 69: "computational" -> "compuationally"
- line 104: "ditribution" -> "distribution"
- line 170 "convergency" -> "convergence"
- line 215 "tunned" -> "tuned"

## Originality
I've not worked in the area of Gaussian graphical models, nor in sparse estimation of precision/covariance matrices, so it's difficult for me to judge how original or novel the paper is. My sense is that the method itself is  built from standard pieces and is not very original, but the theory that establishes the method's consistency and convergence rates is non-trivial and new.

## Significance
While this paper focuses on the particular applied setting of inferring gene networks from RNA sequence data, the method is fairly general to any setting in which multivariate count data arises. The method is simple, easy to understand, and seems easy to implement, so I expect it will be popular among practitioners. I would not be surprised if similar methods are already in use, and thus the new theoretic guarantees that this paper contributes will likely make the proposed method popular in practice.

---

> ### Author Response · Authors · 2022-08-02
> **Response to Reviewer CQBu**
>
> Thank you for your positive feedback and helpful comments. We corrected typos in the manuscript.
>
> For quality:
> >"One drawback is that only the simple case of a banded covariance matrix is given as an example to compare visually the results of the proposed method to the baselines. Perhaps the authors can add more visual inspections to the appendix."
>
> Thank you for your suggestion. We added visual inspections for other graph types in Supplemental Material.
>
> For originality:
> >"My sense is that the method itself is built from standard pieces and is not very original, but the theory that establishes the method's consistency and convergence rates is non-trivial and new."
>
> Thanks for your positive comments. Although the computational method is built from standard pieces, using the maximum marginal log-likelihood estimation (MMLE) to estimate the precision matrix is novel. To our knowledge, due to the complicated form of the PLN's likelihood, all previous methods solve the PLN graphical model using various approximations and it is hard to derive theoretical properties for these approximate methods. Using the MMLE allows us to develop a computational efficient algorithm as well as a nontrivial convergence theory.
>
> For questions:
> >"1. In the experiments section there are 3 instances where PLNet and VPLN seem to achieve the same AUPR, but only PLNet is bolded. Is this just due to rounding?"
>
> Yes. The results in the tables of simulation part are rounded to two decimal places, and the methods that achieve the best performance are bolded. For example, under the scenario of n=500, p=100 and a high dropout rate, PLNet achieves an AUPR of 0.174 while VPLN achieves an AUPR of 0.171.
>
> >"2. What is the relationship of the proposed method to the method given in section 2.4.2 of Fan, Liao, and Liu (2015) [1]?"
>
> The method in section 2.4.2 of Fan, Liao, and Liu (2015) is proposed to estimate the correlation matrix. To guarantee the positive semi-definiteness of the estimated correlation matrix, given an initial estimation of the correlation matrix, this method tries to find a positive semi-definite matrix that is closest to the initial correlation matrix estimation under the Frobenius norm. Our method is to estimate the precision matrix. To estimate the precision matrix, we need an estimation of the covariance matrix that is positive semi-definite. The MMLE can only give us a consistent estimation of the covariance matrix, but cannot guarantee the positive semi-definiteness. Thus, we also try to find a positive semi-definite matrix that is closest to the initial MMLE. However, here we used the element-wise infinity norm to guarantee the element-wise convergence rate of the obtained covariance matrix estimator (see Theorem 1 in the manuscript for details).

---

### Official Review · Reviewer_zkNW · 2022-07-11

**Rating:** 6
**Confidence:** 4
**Soundness:** 3 good
**Presentation:** 2 fair
**Contribution:** 3 good

**Summary:**

This paper proposes PLNet, a two-step approach to estimate the precision matrix of the latent normal distribution which represents the graph of count data. PLNet first considers the Poisson log-normal (PLN) graphical model and utilizes the maximum marginal likelihood estimator (MMLE) to estimate the latent covariance matrix. In the second step, PLNet estimates the sparse precision matrix by minimizing the lasso penalized D-trace loss. The authors also prove that the resulting estimator is a consistent estimator of the precision matrix in high dimensional settings under mild conditions. Extensive experiments, especially simulation experiments, demonstrate that PLNet outperforms baseline methods in both accuracy and efficiency.

**Questions:**

1. The reviewer wonders if it is possible to incorporate the gene regulatory network information in the modeling process to improve the estimation performance?

2. In scRNA-seq data, sometimes gene expression of some genes is not detected. The reviewer wonders if the authors consider using a probability describing whether a gene's expression is detected to better approximate the real data.

**Limitations:**

Authors have discussed the Limitations and societal impacts in the Conclusion Section.


**Strengths And Weaknesses:**

Strengths:

1. Instead of using Gaussian distribution, the authors leverage multivariate Poisson log-normal (PLN) distribution to estimate the graphical models of count data, which handles the over-dispersion problem and is more likely to approximate the distribution of count data like scRNA-seq data.

2. PLNet is computationally efficient as it avoids computing high-dimensional numeric integration when computing the log-likelihood of the multivariate PLN model, and applying the D-trace method to estimate the precision matrix is more efficient.

3. The authors provide the theoretical analysis of the estimator in the high-dimensional setting and prove that the estimator is a consistent estimator of the precision matrix under mild conditions.

Weaknesses:

1. It seems that the authors did not mention how to obtain the mean vector of the latent normal distribution in the PLNet. It would be great if the authors could discuss it.

2. Sometimes in scRNA-seq data, the number of genes (i.e., $p$) may be large, and thus the reviewer wonders if the estimator is still a consistent estimator of the precision matrix in this case?

3. Although extensive simulation results have been presented in this paper, it seems that the real-world experimental results are not very clear. The reviewer thinks that the authors may use some other criteria to quantify the results rather than just using the heatmap and within-between connection ratios.

4. The presentation of experiment results should be improved, and it might be better to use tables to present the results.

---

> ### Author Response · Authors · 2022-08-02
> **Response to Reviewer zkNW**
>
> We would like to thank the reviewer for his/her helpful comments and suggestions.
>
> For weaknesses:
> >"1. It seems that the authors did not mention how to obtain the mean vector of the latent normal distribution in the PLNet. It would be great if the authors could discuss it."
>
> Sorry for the unclear presentation. The means and variances of the latent normal distribution are simultaneously estimated by maximizing the marginal log-likelihood. Recall that $L(\mathbf{Y}\_{\cdot j},\mathbf{S};\mu,\sigma)$ is the  marginal log-likelihood of $Y_{ij}$ ($i=1,\cdots,n$), the mean $\mu_j^*$ and the variance $\sigma_{jj}^*$ are estimated simultaneously by solving the following optimization problem,
> $$(\tilde{\mu}\_j,\tilde{\sigma}\_{jj})=\arg\max_{\mu,\ \sigma}\log L(\mathbf{Y}\_{\cdot j},\mathbf{S};\mu,\sigma). $$
> We also use a Newton-Raphson algorithm to maximize the marginal log-likelihood with initial values setting as the moment estimator of $\mu_j^*$ and $\sigma_{jj}^*$. This discussion was added in the manuscript.
>
> >"2. Sometimes in scRNA-seq data, the number of genes (i.e., $p$) may be large, and thus the reviewer wonders if the estimator is still a consistent estimator of the precision matrix in this case?"
>
> Yes, the estimator is consistent even if $p$ is very large. More precisely, Theorem 2 and 3 in the manuscript guarantee that the proposed estimator is consistent even if $p$ also tends to infinity as the sample size $n$ tends to infinity, but $p$ should tend to infinity not faster than the exponential of $n$.
>
> >"3. Although extensive simulation results have been presented in this paper, it seems that the real-world experimental results are not very clear. The reviewer thinks that the authors may use some other criteria to quantify the results rather than just using the heatmap and within-between connection ratios."
>
> Following the reviewer's suggestion, we use another criterion to quantify the real-world experimental results. Using a known regulatory network database obtained from ChIP-seq experiments (the hTFtarget database), we construct a silver standard network and compare the detected networks by different algorithms with the silver standard network (Table 4 in the revised paper). We tune the tuning parameters of different algorithms such that their estimated networks have the same edge densities (the percentage of the non-zero edges). We find that PLNet detects more known regulatory relationships than other algorithms at the same edge densities, suggesting that PLNet has a higher true discovery rate than other algorithms.
>
> >"4. The presentation of experiment results should be improved, and it might be better to use tables to present the results."
>
> We add a table to show the quantification results of the real-world benchmark analysis in the revision (Table 4 or the table below). Most of results in the manuscript are presented with tables, and two pictures are used to show some results vividly.
> | Density |0.01| 0.02| 0.03| 0.04| 0.05 | 0.06| 0.07 | 0.08 | 0.09 | 0.10 |
> |-----------| ----- |----- |-----|-----|-----|-----|-----|-----|-----|-----|
> | __PLNet__| 8 |16 |23| 35 |41| 44 |62 |73| 81| 92|
> | __VPLN__ | 2| 5| 7| 12| 20| 27| 36| 48| 62| 62|
>
> For questions:
> >"1. The reviewer wonders if it is possible to incorporate the gene regulatory network information in the modeling process to improve the estimation performance?"
>
> This is a great suggestion. PLNet can incorporate the prior information about the gene regulatory relationship to improve the performance of network inference. More specifically, let $E$ be the set of gene pairs that cannot have direct interactions from the prior information. We can incorporate the prior information $E$ by considering the following constrained optimization problem,
> $$
> \widehat{\Theta}=\arg\min_{\Theta\succeq 0} \lbrace \frac{1}{2}{\rm tr}\left( \widehat{\Sigma}\Theta^2\right) -{\rm tr}\left( \Theta \right) +\lambda_n \left\|\Theta\right\|_\{1,\text{off}}: {\Theta\_{ij}} = 0  \ \mbox{for} \  \(i,j\) \in E \rbrace.
> $$
>
> A similar algorithm can be used to solve this optimization problem. This was added in the revision of the manuscript. We will also add this in the implementation of PLNet.
>
> >"2. In scRNA-seq data, sometimes gene expression of some genes is not detected. The reviewer wonders if the authors consider using a probability describing whether a gene's expression is detected to better approximate the real data."
>
> We could use the zero-inflated Poisson log-normal model to address this problem. However, the extra zero-inflated part makes it more difficult to develop convergence theory and we will consider this in the future.
>
> In addition, the PLN model itself partly considers this problem. In the PLN model, the observed expression $Y_{ij}$ follows a Poisson distribution conditional on the true expression $X_{ij}$ of the $j$th gene in the $i$th cell, and $Y_{ij}$ always has a positive probability to be $0$ even if the true expression level $X_{ij}>0$.

---

### Author Response · Authors · 2022-08-09
**Thank you very much for your thorough reviews!**

Thank you for the comprehensive reviews and thoughtful comments. We are delighted that all reviewers appreciated the novelty and the significance of the paper and gave very positive comments.

We are excited about the recognition that “the authors' proposed algorithm (as well as the theoretical analysis of it) is novel" and that "the experiments are reported with sufficient detail".  We are thrilled that the reviewers can see that “the proposed method is grounded in an applied problem”, “methods that are able to operate on raw counts are crucial” and "the authors tackle a high-impact problem". We are also pleased that the reviewers found that "This paper is very well-written and clear" and "the writing in the manuscript was clear and I generally found the paper to be well-structured".

We appreciated the helpful comments and suggestions from reviewers to help us improve our paper. In the revision, we corrected the typos, supplemented the estimation method of the mean vector, added the estimation method with prior network information, and included another criterion for real-world data analysis. We also provided more figures and additional codes in the Supplemental Material and more discussions for possible future work according to the suggestions. We responded to each reviewer separately below. Please let us know if you have additional questions or comments!

---

### Meta-Review · Area_Chair_1MSv · 2022-08-26

**Recommendation:** Accept
**Confidence:** Less certain

**Metareview:**

This paper addresses the problem of identifying a network of interacting entities with applications to genetic regulatory networks. The approach makes use of a Poisson log-normal graphical model structure for count data. The paper shows that the approach has desirable statistical properties such as consistency of the precision matrix estimate. The paper compares the performance to some other methods on various graph structures and presents results on a single-cell RNA sequencing data set.

The reviewers generally found merit in the approach. The clarity of the paper was judged by the reviewers to be good. The suggestion from one reviewer that prior information about the gene regulatory network may be incorporated was welcomed by the authors, though this addition may have an impact on the consistency results. The rate of convergence was found to be root-n and while the practical applicability of that rate for relatively expensive samples such as scRNA-seq is not clear, the method is applicable to a wide range of problems where a sample size that approaches the numbers simulation examples is more feasible. I would strongly encourage the authors to discuss potential limitations including: (1) the interaction between the expense of scRNA-seq data collection and the asymptotics of the consistency results, and (2) the degree to which model misspecification affect scientific conclusions based on parametric inference.

**Award:**

No

---

### Decision · Program_Chairs · 2022-09-14

Accept